# *GmWRKY81* Encoding a WRKY Transcription Factor Enhances Aluminum Tolerance in Soybean

**DOI:** 10.3390/ijms23126518

**Published:** 2022-06-10

**Authors:** Wenjiao Shu, Qianghua Zhou, Peiqi Xian, Yanbo Cheng, Tengxiang Lian, Qibin Ma, Yonggang Zhou, Haiyan Li, Hai Nian, Zhandong Cai

**Affiliations:** 1The State Key Laboratory for Conservation and Utilization of Subtropical Agro-Bioresources, South China Agricultural University, Guangzhou 510642, China; 13632106524@163.com (W.S.); zhouqh20010504@163.com (Q.Z.); pqxian@stu.scau.edu.cn (P.X.); ybcheng@scau.edu.cn (Y.C.); liantx@scau.edu.cn (T.L.); maqibin@scau.edu.cn (Q.M.); 2Guangdong Laboratory for Lingnan Modern Agriculture, Guangzhou 510642, China; 3The Key Laboratory of Plant Molecular Breeding of Guangdong Province, College of Agriculture, South China Agricultural University, Guangzhou 510642, China; 4The Guangdong Subcenter of the National Center for Soybean Improvement, College of Agriculture, South China Agricultural University, Guangzhou 510642, China; 5Hainan Yazhou Bay Seed Laboratory, Sanya Nanfan Research Institute of Hainan University, Sanya 572025, China; ygzhou@hainanu.edu.cn (Y.Z.); hyli@hainanu.edu.cn (H.L.)

**Keywords:** soybean, acid aluminum tolerance, *GmWRKY81*, RNA-seq

## Abstract

Aluminum (Al) toxicity is an essential factor that adversely limits soybean (*Glycine max* (L.) Merr.) growth in acid soils. WRKY transcription factors play important roles in soybean responses to abiotic stresses. Here, *GmWRKY81* was screened from genes that were differentially expressed under Al treatment in Al-tolerant soybean Baxi10 and Al-sensitive soybean Bendi2. We found that *GmWRKY81* was significantly induced by 20 μM AlCl_3_ and upregulated by AlCl_3_ treatment for 2 h. In different tissues, the expression of *GmWRKY81* was differentially induced. In 0–1 cm root tips, the expression of *GmWRKY81* was induced to the highest level. The overexpression of *GmWRKY81* in soybean resulted in higher relative root elongation, root weight, depth, root length, volume, number of root tips and peroxidase activity but lower root average diameter, malonaldehyde and H_2_O_2_ contents, indicating enhanced Al tolerance. Moreover, RNA-seq identified 205 upregulated and 108 downregulated genes in *GmWRKY81* transgenic lines. Fifteen of these genes that were differentially expressed in both AlCl_3_-treated and *GmWRKY81*-overexpressing soybean had the W-box element, which can bind to the upstream-conserved WRKY domain. Overall, the combined functional analysis indicates that *GmWRKY81* may improve soybean Al tolerance by regulating downstream genes participating in Al^3+^ transport, organic acid secretion and antioxidant reactions.

## 1. Introduction

Acid soils (pH < 5.5) are mainly distributed in tropical and subtropical regions worldwide, and nearly 50% of potential arable land is acidic, including about 38% of southeast Asian farmland, 31% of Latin American farmland and 20% of arable land in North America, East Asia and sub-Saharan Africa, which are negatively influenced by aluminum (Al) [1,2]. Almost 35% of soybean cultivation areas in the world involving acid soils can also cause Al toxicity to soybean and lead to yield loss [3]. It has been reported that the content of Al^3+^ in acid soils reaches up to 750 mg kg^−1^, causing Al toxicity in plants [4]. As one of the main factors limiting plant growth, Al^3+^ released from acid soils will inhibit root growth, especially the distal transition zone (DTZ) of the root apex, thereby reducing the absorption of nutrients and water [5]. Positively charged Al^3+^ can bind with negatively charged pectin, resulting in the destruction of the cell wall [6]. Likewise, Al^3+^ can bind to carboxyl and phosphate groups, causing destruction of the plasma membrane [7]. Excessive Al^3+^ accumulation in cells affects the normal function of mitochondria, promoting the production of reactive oxygen species (ROS) and finally inhibiting cell growth [8,9]. In addition, Al^3+^ can reduce the absorption of other positive ions, such as Mg^2+^, K^+^ and NH_4_^+^, as well as promote the absorption of negative ions, such as NO_3_^−^ and PO_4_^3-^, resulting in nutritional imbalance.

To resist Al toxicity, plants developed two main mechanisms during their evolution: the Al exclusion mechanism and the Al tolerance mechanism. The former prevents Al from entering the roots by releasing organic acids such as malate, citrate and oxalate to chelate Al^3+^. The latter sequesters Al in plant organelles such as vacuoles after Al^3+^ enters cells [10,11,12]. Various strategies have been proposed on the basis of the two major mechanisms, for example, chelating Al^3+^ by virtue of organic acid secretion and synthesis, increasing the pH around roots and fixing Al^3+^ by the cell wall to prevent Al^3+^ from entering cells, and increasing antioxidant enzyme activity to enhance Al tolerance [13,14]. Notably, studies on transcription factors (TFs) regulating abiotic stress tolerance in plants have been widely undertaken, commonly focusing on TF families such as ART, MYB and WRKY from modern or important plants. For instance, in indica rice, the C_2_H_2_-type zinc finger TF ART1 confers a higher Al tolerance by regulating 19 downstream genes, and another STOP1 in the same TF family in *Arabidopsis* can activate the expression of the Al resistance gene *AtALMT1* to resist Al toxicity [15,16]. In addition, recent studies have also demonstrated that WRKY transcription factors play a vital role in alleviating biotic and abiotic stresses [17].

WRKY proteins are characterized by a highly conserved WRKY domain with approximately 60 amino acids, of which the N-terminal is the WRKYGQK amino acid sequence and the C-terminal is a C_2_-H_2_ zinc finger motif (CX_4–5_CX_22–23_HXH) or a C_2_-HC zinc finger motif (CX_7_CX_23_HXC) [18]. The conserved domain specifically binds to the DNA sequence TTGACC/T, also named the W-box element, to regulate the expression of downstream genes [19]. In addition, WRKY proteins can be divided into three groups based on the number of WRKYGQK domains and the category of zinc finger motifs. For instance, Group I consists of two WRKYGQK domains and a C_2_H_2_ zinc finger motif. Group II includes a single WRKYGQK domain and a C_2_H_2_ zinc finger motif; furthermore, this group can be divided into five subgroups, denoted IIa-IIe, depending on amino acid differences. Group III encompasses a single WRKYGQK domain and a C_2_-HC zinc finger motif [20]. The first WRKY member was discovered in sweet potato and named SPF1 (sweet potato factor-1) [21]. In soybean, 188 WRKY members are distributed on 20 chromosomes, of which chromosome 6 is the most widely distributed, with 16 WRKY members, accounting for 8.51% [22]. WRKY TFs not only participate in positive regulation but also engage in the negative regulation of abiotic stresses. Under drought stress, GmWRKY27 inhibits the expression of a negative stress tolerance factor named *GmNAC29* by binding to the W-box element and interacts with GmMYB174, which suppresses the expression of *GmNAC29*, thus jointly enhancing drought stress tolerance [23]. In addition, GmWRKY54 binds to the promoter regions of *PYL8*, *SRK2A*, *CIPK11* and *CPK3* and participates in abscisic acid (ABA) and Ca^2+^ signaling pathways, thus improving the drought tolerance of soybean [24]. Under salt stress, overexpression of *GmWRKY12* decreased the content of malondialdehyde (MDA), a product of harmful membrane lipid peroxidation. This shows that *GmWRKY12* can enhance soybean salt tolerance by engaging in antioxidant pathways [25]. However, GmWRKY46 negatively regulates soybean tolerance to phosphorus (P) deficiency by regulating the expression of a number of P-responsive genes, such as *GmPht1;4*, *GmPTF1*, *GmACP1* and *GmPAP21*. RNA interference of *GmWRKY46* results in increased proliferation, elongation and P absorption efficiency of hairy roots [26]. Though many studies have proved the functions of WRKY TFs in the abiotic stress response in soybean, there is still a lack of ample studies focusing on the function and the molecular mechanism of soybean WRKY TFs in the Al stress response.

Soybean is subjected to many abiotic stresses during growth and development, especially Al stress in acid soils, which severely limits the production of soybean and seriously threatens food security [27,28]. In addition, soybean is an important resource for protein and oil for humans worldwide. Therefore, it is necessary to develop effective tactics against Al toxicity in acid soils. In the present study, the *GmWRKY81* gene was selected as a candidate Al-tolerant gene because it was significantly upregulated by Al in both Al-tolerant soybean Baxi10 (BX10) and Al-sensitive soybean Bendi2 (BD2) [29]. We found that *GmWRKY81* is a member of the WRKY family, which plays an important role in plants responding to various stresses; thus, we attempted to verify the function of soybean *GmWRKY81* in Al resistance and further analyze the molecular mechanism of *GmWRKY81* in regulating Al tolerance of soybean.

## 2. Results

### 2.1. Expression Patterns of GmWRKY81

The expression profiles of *GmWRKY81* in response to Al stress were examined by qRT–PCR. *GmWRKY81* transcript levels were significantly enhanced by 20–100 μM AlCl_3_ treatment for 6 h, peaking at 50 μM AlCl_3_ treatment (upregulated approximately 12-fold compared with the expression of *GmWRKY81* without Al treatment), followed by a decline to approximately 5-fold in 100 μM AlCl_3_ treatment (Figure 1A). Al-induced *GmWRKY81* expression was examined in a time-dependent manner. *GmWRKY81* transcript levels were markedly enhanced by 50 μM AlCl_3_ treatment for 2–24 h, reaching the peak value after 6 h of treatment (upregulated approximately 11-fold compared with the expression of *GmWRKY81* without Al treatment), followed by a decline to approximately 3-fold after 24 h of treatment (Figure 1B). The expression of *GmWRKY81* in soybean roots, stems and leaves was also examined. Regardless of whether Al treatment was applied, *GmWRKY81* was mainly expressed in roots and leaves. However, only in roots were *GmWRKY81* transcript levels greatly enhanced after Al treatment, especially in 0–1 cm root tips (upregulated approximately 11-fold compared with the expression of *GmWRKY81* in 0–1 cm root tips without Al treatment) (Figure 1C).

### 2.2. Cloning and Characteristic Analysis of GmWRKY81

Full-length *GmWRKY81* (*Glyma.04G061400*) was isolated from the Al-tolerant soybean variety BX10 according to putative sequence information from Phytozome v13 (https://phytozome-next.jgi.doe.gov/, accessed on 4 February 2022). The coding sequence (CDS) of *GmWRKY81* was 663 bp (Appendix A), encoding a predicted protein of 220 amino acids with an estimated molecular weight of 24.926 kDa and an isoelectric point of 8.46. A complete phylogenetic tree constructed with all 188 WRKY proteins in soybean showed that GmWRKY81 belongs to the IIa subfamily (Appendix A). In addition, AtWRKY40, AtWRKY18 and AtWRKY60 in *Arabidopsis thaliana* also belong to this subfamily, sharing high homology with GmWRKY81 (Figure 2A). Protein sequence alignment demonstrated that GmWRKY81 contains a conserved WRKY domain that includes the amino acid sequence “WRKYGQK” and a C_2_H_2_ zinc finger structure (Figure 2B). Cell-Ploc 2.0 (http://www.csbio.sjtu.edu.cn/bioinf/Cell-PLoc-2/, accessed on 4 February 2022) was employed to predict the subcellular localization of GmWRKY81 and showed that GmWRKY81 is localized in the cell nucleus. To further confirm the predicted result, the CDS of *GmWRKY81,* except for its stop codon, was linked to the 5′-terminus of green fluorescent protein (GFP) in pCAMBIA1302, promoted by the cauliflower mosaic virus (CaMV) 35S promoter. The fusion protein 35S::GmWRKY81-GFP was temporarily expressed in the tobacco lower epidermis to observe the protein location. Laser confocal microscopic observation revealed that green fluorescence only existed in the nucleus, coinciding with the staining position of the nuclear dye DAPI (Figure 2C). This result indicates that GmWRKY81 is located in the cell nucleus.

### 2.3. Overexpression of GmWRKY81 Improves Al Tolerance in Soybean

To explore the function of *GmWRKY81* in Al tolerance, *GmWRKY81* transgenic lines were obtained by transformation of soybean cotyledon nodes with the Al-sensitive wild type (WT), named Guixia1, as the recipient. Three transgenic lines (Line 3, Line 7 and Line 13) were selected from a total of 15 transgenic lines for further phenotypic identification of Al resistance because of their higher expression levels of *GmWRKY81* (Appendix A). The results of short-term hydroponics for 48 h showed that growth inhibition was enhanced as AlCl_3_ concentration increased, but the inhibitory effects in the transgenic lines and WT were not the same (Figure 3A). As shown in Figure 3B, the relative root elongation in the transgenic lines was approximately 30% higher than that in WT under the 25 and 50 μM AlCl_3_ treatments. When the AlCl_3_ concentration increased to 100 μM, the relative root elongation gap narrowed to approximately 15%.

To further explore the role of *GmWRKY81* in alleviating root inhibition, WT and transgenic plants were cultivated in a nutrient solution treated with different concentrations of AlCl_3_ for 10 d, and then root fresh and dried weight measurements, root scanning analysis and root physiological index determination were carried out. As shown in Figure 4A, the roots of WT and transgenic lines treated with 20 μM AlCl_3_ were visibly weaker than those treated with 10 μM AlCl_3_, but the roots of transgenic lines were much stronger than those of WT under the two treatments. The total root length, depth, volume and number of root tips showed a downward trend as the AlCl_3_ concentration increased, and the indices of the transgenic lines were noticeably higher than those of WT under the 10 and 20 μM AlCl_3_ treatments (Figure 4B–E). However, the average root diameter was the opposite (Figure 4F). Moreover, both root fresh and dried weight showed a similar change tendency with total root length; nevertheless, the root dried weight of transgenic lines was much higher than that of WT only under the 20 μM AlCl_3_ treatment (Figure 4G,H). In addition, MDA and H_2_O_2_ contents and peroxidase (POD) activity were measured to reflect the effect of *GmWRKY81* on root physiology. Treatment with 20 μM AlCl_3_ led to increases in MDA and H_2_O_2_ contents and POD activity in both WT and transgenic roots. Before AlCl_3_ treatment, only one transgenic line had considerably lower MDA content or H_2_O_2_ content than WT, but when AlCl_3_ was applied, all transgenic lines had evidently lower MDA and H_2_O_2_ contents than WT (Figure 5A,B). POD activity in transgenic Lines 3 and 7 was dramatically higher than that in WT with AlCl_3_ treatment. For transgenic Line 13, POD activity was significantly higher than that of WT only after AlCl_3_ treatment (Figure 5C).

### 2.4. GmWRKY81 Overexpression Results in Differential Transcription of Al-Responsive Genes

To obtain greater insight into the underlying molecular mechanism modulating Al tolerance and stress-responsive genes by *GmWRKY81*, RNA-seq was conducted with WT and transgenic line 13 as experimental materials. After screening by |log2 fold change| > 1.0 and *p* value < 0.05, a total of 313 genes showed different transcript levels between the transgenic lines and WT, of which 108 genes were downregulated and 205 genes were upregulated in *GmWRKY81*-overexpressing lines (Figure 6A and Appendix A). Six differentially expressed genes (DEGs) were selected randomly to validate the RNA-seq results by qRT–PCR. As seen in Figure 6B, the qRT–PCR results were basically in line with the RNA-seq results, indicating that the above DEGs selected by RNA-seq were reasonable.

Based on gene ontology (GO) analysis, a total of 109 GO terms were significantly enriched in DEGs, with the top 20 participating in biological processes such as “response to wounding”, “H_2_O_2_ catabolic process”, “response to oxidative stress” and “nicotianamine biosynthetic process”; taking part in cellular components such as “extracellular region”, “extracellular matrix”, “apoplast” and “cell wall”; and participating in molecular functions including “heme binding”, “peroxidase activity” and “pyruvate decarboxylase activity” (Figure 6D and Appendix A). The Kyoto Encyclopedia of Genes and Genomes (KEGG) analysis found 76 enriched pathways in DEGs, with the top 20 including “arginine and proline metabolism”, “flavonoid biosynthesis”, “fatty acid elongation and degradation” and “biosynthesis of unsaturated fatty acids” (Figure 6E and Appendix A).

Venn diagram analysis showed that 15 genes were shared among the DEGs induced by Al in the roots of the Al-tolerant soybean BX10 and the Al-sensitive soybean BD2 and the overexpression of *GmWRKY81* (Figure 6C). These 15 shared genes all had the W-box element, which can particularly bind to the WRKY transcription factor upstream. In addition, seven genes were notably upregulated and eight genes were prominently downregulated by Al in BX10 and BD2. According to the gene descriptions in the NCBI, three genes are related to metal transport (*Glyma.06G115800*), metal deficiency (*Glyma.03G130600*) and metal tolerance (*Glyma.09G122600*); five genes encode different enzymes, including xanthotoxin 5-hydroxylase CYP82C4 (*Glyma.04G035600*), ferric reduction oxidase (*Glyma.07G067700*), nicotianamine synthase (*Glyma.08G175400* and *Glyma.15G251300*) and cationic peroxidase (*Glyma.18G211000*) (Appendix A).

## 3. Discussion

Al is the most abundant metal in the Earth’s crust, and it severely limits crop growth and production in acid soils. Numerous tactics to cope with metal stress have been proposed, among which transcription-factor-assisted responsive mechanisms have great potential. The WRKY transcription factor is typically involved in dealing with metal tolerance. For example, OsWRKY22 was reported to promote Al tolerance by activating *OsFRDL_4_* expression [30]; AtWRKY47 was observed to confer Al tolerance by regulating cell-wall-modifying genes [31]; and AtWRKY46 acts as a transcriptional repressor of *AtALMT1* to negatively regulate Al resistance [32]. However, reports on the role of WRKY TFs in alleviating Al toxicity in soybean are still rare. With the development of the RNA-seq technique, more candidate genes have been identified from gene expression profiles under stress conditions. Here, we screened and cloned the candidate gene *GmWRKY81,* which encodes a WRKY protein that participates in the Al stress response from BX10 and BD2, based on the RNA-seq results of previous research [29] and further explored the molecular mechanism underlying the regulation of aluminum tolerance in soybean.

Research has discerned that members of the WRKY family specifically bind to their downstream W-box elements containing TTGACC/T to regulate the expression of downstream genes as transcription factors, thus protecting plants against various stresses [33]. Consistently, in our study, the protein sequence alignment and the phylogenetic tree analysis of GmWRKY81 and other WRKY proteins showed that GmWRKY81 belongs to the IIa subfamily, which contains a WRKYGQK protein motif and C_2_H_2_ zinc finger structure. Genes with relatively high homology to *GmWRKY81,* such as *Glyma.06G061900* (*GmWRKY17*) and *Glyma.17G222300* (*GmWRKY30*), participate in the response of soybean to cadmium (Cd) stress [17] and iron deficiency stress [34], respectively. Moreover, in *Arabidopsis*, homologous genes such as *At2G25000* (*AtWRKY60*), *At4G31800* (*AtWRKY18*) and *At1G80840* (*AtWRKY40*) are involved in the response to abiotic stresses in plants, forming a highly coordinated regulatory network to jointly regulate the expression of downstream genes, thus enhancing plant resistance to abiotic stresses [35]. The subcellular location showed that GmWRKY81 is located in the nucleus (Figure 2). Therefore, we speculate that GmWRKY81 may take part in responding to Al stress, a kind of abiotic stress.

The expression pattern of *GmWRKY81* exposed to Al showed that *GmWRKY81* was significantly induced by 20 μM AlCl_3_ and markedly upregulated by AlCl_3_ treatment for 2 h. In different tissues, the expression levels of *GmWRKY81* were differentially induced. In 0–1 cm root tips, the expression of *GmWRKY81* was induced the most (Figure 1). This implies that *GmWRKY81* is indeed activated by Al stress. This is consistent with previous reports revealing the existence of Al-induced WRKY proteins [31]. Overexpression of *GmWRKY81* in short- and long-term AlCl_3_-treated soybean resulted in a better performance in regard to root elongation and morphology. Moreover, the growth inhibition of roots was decreased in overexpressed lines relative to WT (Figure 3 and Figure 4). These results suggest that *GmWRKY81* may render plants more tolerant to Al than WT plants. Similar observations were also made for *OsWRKY22* [30]. Recent research revealed that Al stress can directly exert negative effects on the normal function of plant mitochondria, which leads to the accumulation of numerous H_2_O_2_ molecules and thus causes lipid peroxidation [8]. In addition, MDA is an indicator of oxidative damage caused by stress [36]. POD is a defensive enzyme that functions in ROS scavenging (such as H_2_O_2_) to protect plant cells from damage under stress. In our study, compared to WT, the roots of transgenic plants suffered from lower MDA and H_2_O_2_ levels, while their POD activity was significantly increased (Figure 5), reflecting that the enhanced ability of the antioxidant system might be tightly related to increased resistance to Al toxicity, which has a perfect correspondence to alleviated root inhibition in transgenic plants. These results further suggest that *GmWRKY81* is highly associated with soybean resistance to Al.

GO enrichment analysis and KEGG enrichment analysis were performed to explore the biological functions of 313 DEGs. First, the antioxidant reaction process regulated by *GmWRKY81* is of great importance in Al tolerance because numerous DEGs were enriched in the H_2_O_2_ catabolic process, oxidative stress response process, flavonoid biosynthesis, proline metabolism and biosynthesis of unsaturated fatty acids. Notably, reports have shown that flavonoids and proline are associated with ROS clearance [37,38], and MDA is produced from unsaturated fatty acids by lipid peroxidation [39], which further confirms our hypothesis that *GmWRKY81* improves the Al tolerance of soybean chiefly by regulating the expression of related genes in the antioxidant system. Second, the secretion of organic acids affected by *GmWRKY81* is also a vital way to alleviate Al toxicity, which can be deduced from the GO term “pyruvate decarboxylase activity”, because research has demonstrated that malate, one of the organic acids, is the final product of pyruvate decarboxylation in the tricarboxylic acid cycle (TCA) (Figure 6D,E). This is consistent with previous studies showing that Al^3+^ is detoxified by the secretion of organic acids such as malic, citric and oxalic acids from roots [27,40].

Furthermore, we discovered that the expression of 15 shared DEGs changed significantly not only under Al stress but also in the *GmWRKY81*-overexpressing line, revealing the possibility that GmWRKY81 might regulate the expression of these genes on account of their mutual W-box element to participate in different physiological processes and ultimately improve the Al tolerance of soybean (Figure 6C, Appendix A). For instance, *Glyma.03G130600* was significantly downregulated by Al in Al-tolerant and -sensitive varieties. In addition, overexpression of *GmWRKY81* also suppressed *Glyma.03G130600*. According to previous research, GmbHLH300 encoded by *Glyma.03G130600* was reported to be induced by iron deficiency to maintain iron homeostasis [41]. Since Al and iron are both metals, we predict that GmWRKY81 participates in maintaining Al homeostasis by depressing *Glyma.03G130600* when Al is excessive. Likewise, *Glyma.06G115800* was downregulated by Al, and GmNRAMP7 encoded by *Glyma.06G115800* was also reported to be enhanced by Fe starvation [42], so *Glyma.06G115800* may be involved in maintaining Al homeostasis as well. Additionally, OsNrat1, which belongs to the Nramp family, was reported to be an Al^3+^ transporter on the plasma membrane [43]. Based on this, Glyma.06G115800 is likely to maintain Al balance by transporting Al into cells or out of cells. Previous research has demonstrated that *Glyma.15G251300* encoding nicotianamine synthase NAS1 plays a central role in long-distance Fe transport [44]. Though Al transport was not involved in the research, other research has proved that mugineic acid (MA), with nicotianamine (NA) as the important intermediate in its production, has the ability to bind to Al [45,46]. Thus, we speculate that *Glyma.15G251300* can also take part in Al transport in an indirect way. The same function is possible for *Glyma.08G175400*, which encodes nicotianamine synthase as well. The above analysis led us to assume that GmWRKY81 improves the Al tolerance of soybean by regulating genes involved in transporting Al and maintaining Al homeostasis. Interestingly, *Glyma.13G333100* encodes tonoplast intrinsic protein (TIP), which has been demonstrated to channel H_2_O_2_ in *Arabidopsis* [47]. Based on this, we hypothesize that TIP transports excess H_2_O_2_ into vacuoles to isolate it from other sensitive cell components. Consistent with enrichment analysis results, GmWRKY81 might regulate the expression of genes promoting organic acid secretion from roots for chelating Al^3+^ to form a nontoxic Al^3+^–organic acid complex since the homologous gene of *Glyma.19G173800* in *Arabidopsis*, named *At2G39510*, was reported as possibly related to organic acid transport [48]. Meanwhile, expression changes in *Glyma.07G067700*, encoding ferric reduction oxidase, and *Glyma.18G211000*, encoding cationic peroxidase [49], are in high accordance with alterations in physiological indices, indicating that GmWRKY81 can alleviate Al toxicity in soybean via ROS scavenging by upregulating genes related to the antioxidant reaction. Taken together, in the current study, we cloned and identified *GmWRKY81* functioning in the regulation of downstream Al tolerance-related genes. We also revealed the molecular mechanisms underlying Al tolerance as follows: the GmWRKY81 transcription factor mainly activates the antioxidant enzyme system; promotes organic acid secretion; and adjusts the balance of Al^3+^ absorption, transportation, distribution and excretion to enhance Al tolerance in soybean. Moreover, our study provides directions for future research on other types of stresses with the help of various transcription factors (Figure 7).

## 4. Materials and Methods

### 4.1. Plant Materials and Treatments

For expression pattern analysis, approximately 36 plump seeds of the Al-tolerant soybean BX10 were sown in each pot with sterile vermiculite, and seedlings of similar size after 2 d of germination were precultured in 0.5 mM CaCl_2_ (pH = 4.5) for 24 h. Later, these seedlings were transferred into 1/10 modified Hoagland solution containing 2.5 mM KNO_3_, 2.5 mM Ca(NO_3_)_2_·4H_2_O, 4.57 μM MnCl_2_·4H_2_O, 0.25 mM K_2_SO_4_, 1 mM MgSO_4_·7H_2_O, 0.38 μM ZnSO_4_·7H_2_O, 1.57 μM CuSO_4_·5H_2_O, 0.09 μM (NH_4_)_6_Mo_7_O_24_·4H_2_O, 0.2313 μM H_3_BO_3_ and 0.082 mM Fe-EDTA(Na) with various treatments (pH = 4.5). In the concentration-gradient experiment, seedlings were treated with 0, 10, 20, 30, 50 and 100 μM AlCl_3_ for 6 h, and then 0–2 cm of root tips was quickly frozen in liquid nitrogen and stored at −80 °C. In the time-gradient experiment, seedlings were treated with 50 μM AlCl_3_ for 0, 2, 4, 6, 8, 12 and 24 h, and then 0–2 cm of root tips was quickly frozen in liquid nitrogen and stored at −80 °C. Different treatment concentrations and times were set on the basis of previous research [16,50,51]. In the tissue-dependent experiment, after trifoliate leaves fully unfolded, 50 μM AlCl_3_ was added to nutrient solution without AlCl_3_ as a control, and 0–1 cm of root tips, 1–2 cm of root tips, 2–5 cm of root tips, stems and leaves treated for 6 h were quickly frozen in liquid nitrogen and stored at −80 °C. All of the experiments were performed with three biological replicates. For short-term hydroponic phenotypic identification, seeds of wild-type (WT) (Guixia1, Al-sensitive) and transgenic lines were germinated and precultured in the process mentioned above and then treated in 1/10 modified Hoagland solution with 0, 25, 50 and 100 μM AlCl_3_ for 48 h (pH = 4.5). For long-term hydroponic phenotypic identification, the AlCl_3_ concentrations were changed to 0, 10 and 20 μM, and the treatment time was changed to 10 d. All experiments were performed with three biological replicates. For RNA-seq, 0–2 cm of root tips of WT (Guixia1, Al-sensitive) and transgenic line 13 with the highest expression of *GmWRKY81* precultured in 0.5 mM CaCl_2_ for 24 h were quickly frozen in liquid nitrogen and stored at −80 °C until use. RNA-seq was performed with two biological replicates.

### 4.2. qRT–PCR Analysis of Gene Expression

Total RNA of the abovementioned samples was extracted by RNA isolater Total RNA Extraction Reagent (Vazyme, R401-01). Reverse transcription of purified RNA was performed according to the instructions of HiScript III RT SuperMix for qPCR (+gDNA wiper) (Vazyme, R233-01). qRT–PCR was implemented with *Actin 3* as a reference gene and cDNA as an amplified template on a CFX96 Real-Time system (Bio–Rad, Hercules, USA) according to ChamQ Universal SYBR qPCR Master Mix (Vazyme, Q711). The expression levels of genes were calculated by the 2^−ΔΔCT^ method among three biological replicates to ensure the accuracy of the results [52]. The primers used in the experiment are listed in Appendix A.

### 4.3. Bioinformatics Analysis and RNA-seq

The sequences of the *GmWRKY81* gene and GmWRKY81 protein as well as their homologous sequences were searched in Phytozome v13 (https://phytozome-next.jgi.doe.gov/, accessed on 4 February 2022). The locations of conserved WRKY domains and amino acid sequences were examined in SMART (http://smart.embl-heidelberg.de/, accessed on 4 February 2022) and NCBI (https://www.ncbi.nlm.nih.gov/, accessed on 4 February 2022). MEGA 7 was used to construct a phylogenetic tree by aligning protein sequences with ITOL (https://itol.embl.de/, accessed on 4 February 2022) for further modification. The protein sequence alignment was constructed in GENEDOC. RNA-seq was conducted by LC-Bio company from Hangzhou, China. Total RNA was extracted by RNA isolater Total RNA Extraction Reagent (Vazyme, R401-01). The cDNA libraries were constructed with fragmented mRNA, cDNA synthesized from mRNA and T4 polynucleotide kinase, qualified by an Agilent 2100 Bioanalyzer, quantified by an ABI qPCR System and sequenced by an Illumina Novaseq^TM^ 6000. After filtering, clean reads were mapped to the soybean reference genome (https://www.ncbi.nlm.nih.gov/assembly/GCF_000004515.5, accessed on 2 November 2021). Fragments per kilobase of exon model per million mapped reads (FPKM) presented the gene expression level. Gene ontology (GO) (http://www.geneontology.org, accessed on 2 November 2021) was utilized to enrich differentially expressed genes (DEGs) into various GO terms and calculate the gene numbers in each GO term. Finally, significantly enriched GO terms were selected by the hypergeometric test [14]. Furthermore, the Kyoto Encyclopedia of Genes and Genomes (KEGG) (http://www.kegg.jp/kegg, accessed on 2 November 2021) was used to enrich DEGs into different pathways to learn more biological functions of DEGs.

### 4.4. Subcellular Localization of GmWRKY81

To localize the GmWRKY81 protein, the full-length coding sequence of *GmWRKY81*, except for the stop codon, was fused to pCAMBIA1302 with *Nco* I and *Spe* I as insertion sites to obtain the recombinant plasmid pCAMBIA1302-*GmWRKY81*-GFP, driven by the cauliflower mosaic virus 35S (CaMV 35S) promoter. The recombinant plasmid was transformed into *Agrobacterium tumefaciens* GV3101 and cultured to OD_600_ = 0.6. Later, GV3101 with pCAMBIA1302-*GmWRKY81*-GFP and p19 were mixed at 1:1 and incubated for 3–4 h. Finally, the incubated bacterial mixture was injected into leaves of tobacco (*Nicotiana benthamiana*). The color of the lower epidermal cells infected with GV3101 was observed under a laser confocal microscope after 3 d. The primers used in the experiment are listed in Appendix A.

### 4.5. Vector Construction and Transformation of GmWRKY81 in Soybean

The coding sequence of *GmWRKY81* without a stop codon was fused to the pTF101.1 vector with *Xba* I and *Sac* I as insertion sites to construct a recombinant plasmid promoted by the CaMV 35S promoter. The recombinant plasmid was subsequently transformed into *Agrobacterium* EHA105 and AGL1 by the heat-shock method. Subsequently, *Agrobacterium*-mediated transformation of soybean cotyledon nodes was conducted, and the regenerated plants were preliminarily selected by smearing herbicide and further confirmed by PCR and qRT–PCR identification. Positive transgenic plants were cultivated until homozygous. The primers used in the experiment are listed in Appendix A.

### 4.6. Morphological and Physiological Parameter Measurement under Al Stress

To determine the morphological changes in soybean in response to Al stress, root elongation of WT and transgenic lines was measured after short-term AlCl_3_ treatment. Relative root elongation was calculated from the formula: Relative root elongation (%) = root elongation of different lines in different treatments/root elongation of WT in treatment with 0 μM AlCl_3_ * 100%. For further investigation of the effect of Al on roots, a long-term hydroponic experiment was performed to measure more morphological and physiological parameters under Al stress. After treatment, roots of both WT and transgenic lines were scanned in a root scanner, and the representative root indices, including total root length, depth, number of root tips, volume and average diameter, were analyzed in RhizoVision Explorer software. Root fresh weight was measured immediately, and root dried weight was measured after the root was fully dried until a constant weight was obtained. MDA content was detected by the thiobarbituric acid method (TBA), H_2_O_2_ content was detected by the titanium sulfate colorimetry method, and POD activity was detected by spectrophotometry (referring to the explanatory memorandum of the Keming Biological Kit).

### 4.7. Statistical Analysis

GraphPad Prism 6 was used to calculate the average value and standard deviation. SPSS 21 was applied to identify significant differences by ANOVA (*p* < 0.05).

## Figures and Tables

**Figure 1 ijms-23-06518-f001:**
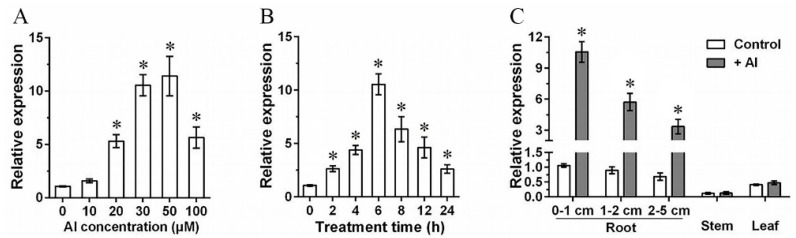
Analysis of *GmWRKY81* gene expression patterns. (**A**) Relative expression of *GmWRKY81* after treatment with 0, 10, 20, 30, 50 and 100 μM AlCl_3_ for 6 h. (**B**) Relative expression of *GmWRKY81* with 50 μM AlCl_3_ treatment for 0, 2, 4, 6, 8, 12 and 24 h. (**C**) Relative expression of *GmWRKY81* in different tissues of plants (roots, stems and leaves) with or without 50 μM AlCl_3_ treatment for 6 h. Asterisks denote significant differences: *p* < 0.05. The data are presented as the means ± SD from three biological replicates.

**Figure 2 ijms-23-06518-f002:**
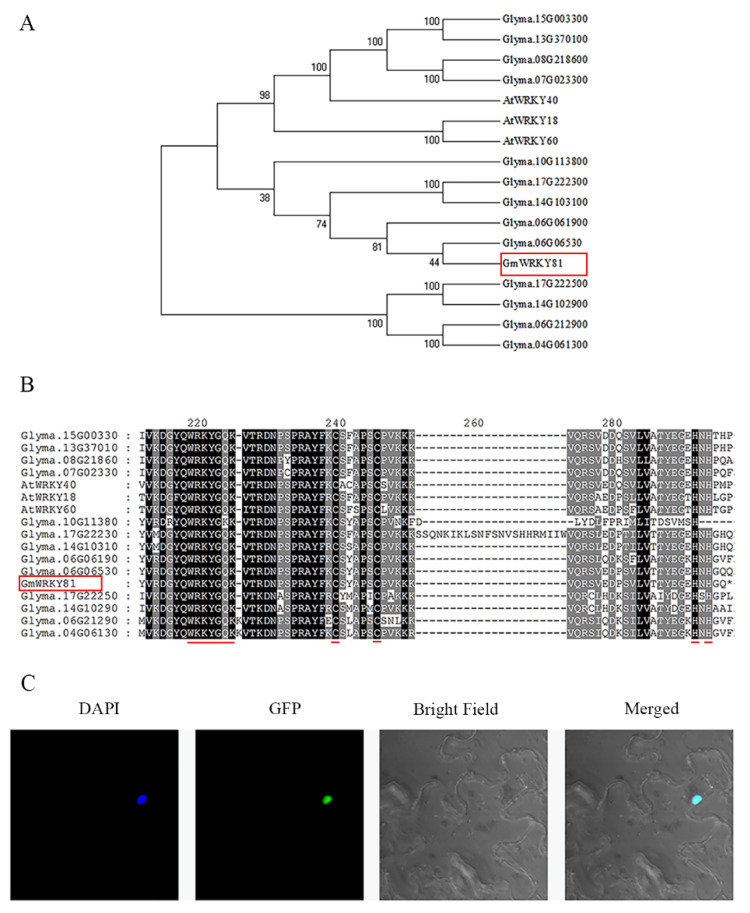
Homology analysis and subcellular localization of GmWRKY81. (**A**) Phylogenetic tree analysis of proteins from IIa subfamily of WRKY family in soybean and *Arabidopsis thaliana*. (**B**) Multiple amino acid sequence alignment of proteins from IIa subfamily of WRKY family in soybean and *Arabidopsis thaliana*. The GmWRKY81 protein is marked with a red rectangle, and the red underline marks the WRKYGQK motif and C_2_H_2_ zinc finger structure. (**C**) The subcellular localization of GmWRKY81 in epidermal cells of tobacco (*N. benthamiana*). DAPI: 4′,6-diamidino-2-phenylindole, a nucleus-specific fluorescence dye. GFP: green fluorescence protein. Merged: the overlapped image of DAPI, GFP and bright field.

**Figure 3 ijms-23-06518-f003:**
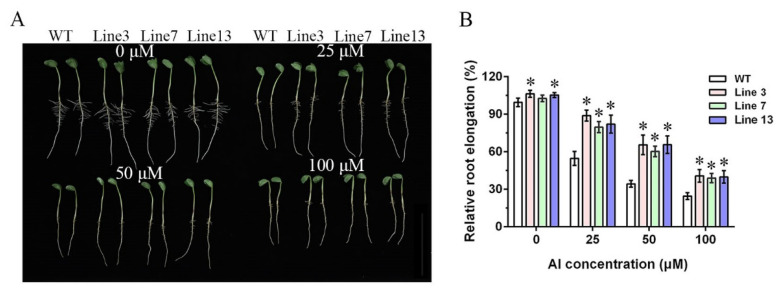
Short-term hydroponic phenotypic identification in wild type (WT) and transgenic lines under different AlCl_3_ treatments. (**A**) Growth performance of WT and transgenic lines treated with 0, 25, 50 and 100 μM AlCl_3_ for 48 h. (**B**) Statistical analysis of relative root elongation in WT and transgenic lines treated with 0, 25, 50 and 100 μM AlCl_3_ for 48 h. Relative root elongation (%) = root elongation of different lines in different treatments/root elongation of WT in treatment with 0 μM AlCl_3_ * 100%. WT: wild type (Guixia1, Al-sensitive). Line 3, Line 7 and Line 13: *GmWRKY81* transgenic lines. Asterisks denote significant differences: *p* < 0.05. The data are presented as the means ± SD from three biological replicates, and each biological replicate included 4 plants.

**Figure 4 ijms-23-06518-f004:**
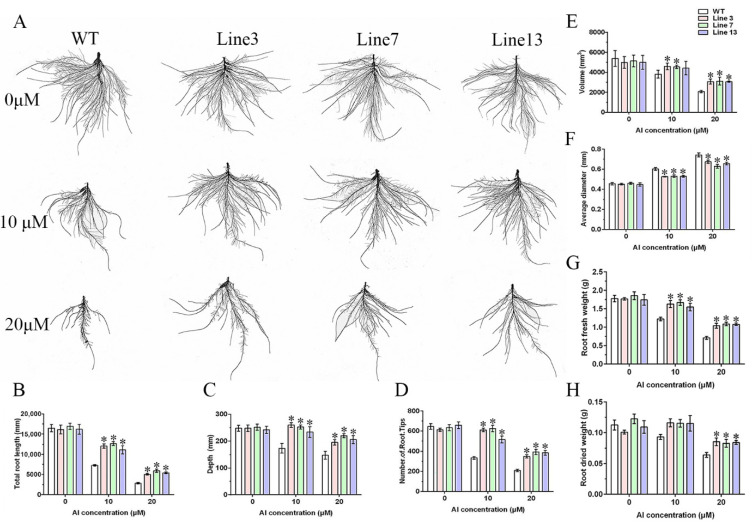
Long-term hydroponic phenotypic identification in WT and transgenic lines under 0, 10 and 20 μM AlCl_3_ treatment. (**A**) Growth performance of WT and transgenic lines treated in long-term hydroponic experiments. Morphological parameters of total root length (**B**), depth (**C**), number of root tips (**D**), volume (**E**), average diameter (**F**), root fresh weight (**G**) and root dried weight (**H**) after 10 d of treatment were analyzed by RhizoVision Explorer software. WT: wild type (Guixia1, Al-sensitive). Line 3, Line 7 and Line 13: *GmWRKY81* transgenic lines. Asterisks denote significant differences: *p* < 0.05. The data are presented as the means ± SD from three biological replicates, and each biological replicate included 4 plants.

**Figure 5 ijms-23-06518-f005:**
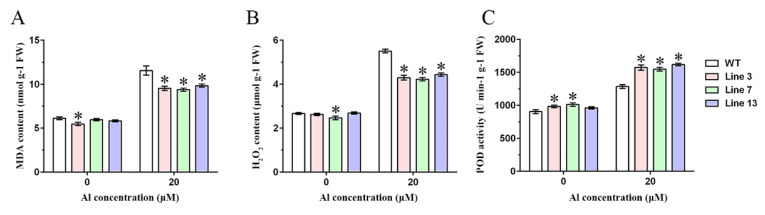
Determination of long-term hydroponic physiological parameters. Malonaldehyde (MDA) content (**A**), H_2_O_2_ content (**B**) and peroxidase (POD) activity (**C**) in roots of WT and transgenic lines treated with 0 and 20 μM AlCl_3_ for 10 d. WT: wild type (Guixia1, Al-sensitive). Line 3, Line 7 and Line 13: *GmWRKY81* transgenic lines. Asterisks denote significant differences: *p* < 0.05. The data are presented as the means ± SD from three biological replicates, and each biological replicate included 4 plants.

**Figure 6 ijms-23-06518-f006:**
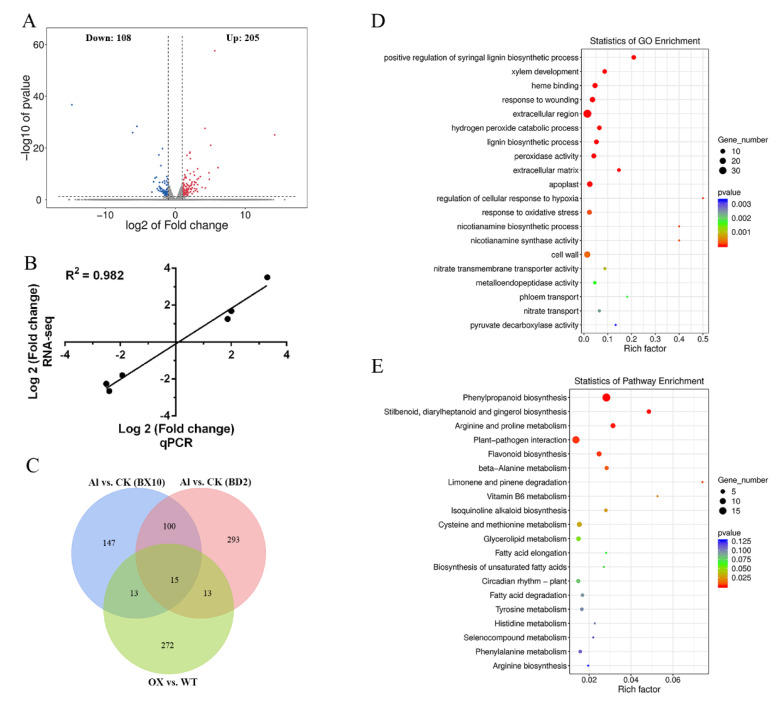
Differentially expressed gene (DEG) identification, gene ontology (GO) enrichment analysis and Kyoto Encyclopedia of Genes and Genomes (KEGG) enrichment analysis. (**A**) Scatter plot of DEGs between transgenic lines and WT. The red dots and blue dots represent up- and downregulated genes, respectively. (**B**) Correlation analysis between qPCR and RNA-seq results. (**C**) Venn plots of DEGs induced by Al in Baxi10 (BX10) (blue), Bendi2 (BD2) (pink) and GmWRKY81-overexpressing lines (green); the numbers represent the quantities of DEGs; fifteen shared DEGs are shown in the overlapping region. (**D**) The result of GO enrichment analysis. The diameter and color depth of each circle are positively correlated with the number of enriched genes and significance level, respectively. (**E**) The result of KEGG enrichment analysis. The diameter and color depth of each circle are positively correlated with the number of enriched genes and significance level, respectively.

**Figure 7 ijms-23-06518-f007:**
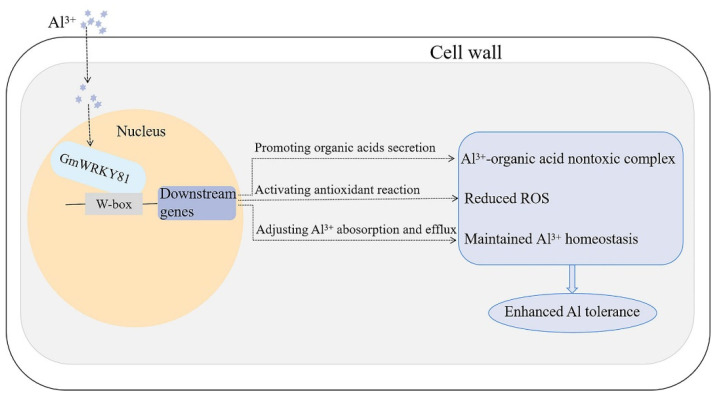
Schematic diagram of GmWRKY81 in regulating soybean Al tolerance. Toxic Al^3+^ in cells activates the expression of *GmWRKY81* and then affects the transcription of downstream genes participating in organic acid secretion, antioxidant reactions and Al^3+^ absorption and efflux processes, thereby accelerating the combination of the nontoxic Al^3+^–organic acid complex, reducing excessive reactive oxygen species (ROS) and maintaining Al^3+^ homeostasis. In these ways, GmWRKY81 confers enhanced Al tolerance to soybean.

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
