# Peer review of "GmWRKY81 Encoding a WRKY Transcription Factor Enhances Aluminum Tolerance in Soybean"

_ijms, 2022, doi:10.3390/ijms23126518_

Round 1

Reviewer 1 Report

WRKY transcription factors that specifically existed in plants have diverse important roles in regulating plant growth and stress responses. Here the authors showed that soybean GmWRKY81 may have a positive role in Al stress resistance by possibly regulating genes associated with antioxidant reactions, organic acid exudation, and metal transport. These results are interesting, and reveal the physiological function of GmWRKY81 in Al tolerance for the first time.

Some comments should be considered:

The identification of transgenic plants by detecting transcription level and/or protein levels should be provided.

The expression patterns of the putative downstream genes of GmWRKY81 under Al treatment and between the two soybean varieties (Al-sensitive and Al-tolerant) should be provided and discussed.

The language of the manuscript should be improved.

The format of plant generic name, e.g., LINE 137: “Arabidopsis thaliana”, LINE 437: “phaseolus vulgaris ” and LINE489: “Cucurbita pepo” should be italicized.

Reviewer 2 Report

This manuscript attempts to describe the role of GmWRKY81 in Aluminum stress tolerance. This is an important abiotic stress for crops, though it’s unclear how important it is for soybean production.  The authors have done a number of good experiments and have generated some good figures. However, it’s unclear to me why the authors chose to investigate the role of GmWRKY81 in Al stress tolerance.  The RNAseq appears to include the over-expression lines, so it wasn’t identified from the RNAseq.  Why was this gene chosen?  What is the GlymaID of GmWRKY81?  Further, the lack of details throughout the manuscript leaves more questions than it answers these questions sow doubt as to what was actually compared and what does it mean.  The authors need to incorporate more references and citations.  Multiple times throughout the manuscript a very detailed claim is made without a reference to substantiate it.  While I appreciate that this is a soybean-focused paper, there have not been a lot of gene specific studies in soybean.  For that reason, soybean studies have to rely on findings from other species, primarily Arabidopsis since it’s a dicot and rice is a monocot.  This manuscript barely mentions the role of the genes discussed in Arabidopsis, which I think would lend a lot of credence to the study.  My issues with the RNAseq aspect of the study are provided in detail below.

Abstract: Don’t use abbreviations in the abstract, unless it’s extremely common (RNAseq is ok). Introducing Al as the abbreviation for aluminium is ok.  All others should be written out and abbreviations introduced in the text of the manuscript.

Pg 2: For an accurate count of soybean TFs, from any family, you should really be using the SoyDB (Wang et al., 2010), citation.

Pg 2:  There is a lack of papers examining the specific role of ANY gene in soybean, not just WRKY TFs under Al stress.  With something that specific, of course there are very few papers.

Ln 83: Explain that MDA is produced when there is an excess of free radicals and hence increased MDA levels indicate oxidative stress.  This is important since you bring up MDA later, but without context of why it’s important.

Ln 89:  Was soybean infected with multiple fungi, or just P. pachyrhizi isolates?  If just rust isolates, say so.  You can indicate that P. pachyrhizi is a fungal spore, but as written this sentence is convoluted and confusing.

Ln 100- There should be a reference for yield lost to Al stress.  Acidic soils result in all kinds of stresses, what is the impact on yield for Al stress (and I think you’re discussing Al toxicity, not deficiency).

Ln 113:  Measuring GmWRKY81 expression in response to Al levels. This is great. However, in acidic soils, what would be the upper and lower-limits of Al availability.  Does 50uM of Al correspond to levels in acidic soil conditions?  Is it too little, too much?  On Ln 115, you state GmWRKY81 is upregulated 12 fold compared to the control, but what is the control?

Ln 138: AtWRKY40 has been implicated in many stresses, so it’s not surprising to see it implicated in Al toxicity.

Pg 4: Figure 2 is mis-placed and the figure legend is present, but not adjacent to the figure itself. Please fix.

            Figure 2: Previous statements say there are 188 WRKY TFs in soybean. This figure indicates that there are only 16 WRKY TFs in Arabidopsis.  Recent studies indicate Arabidopsis has >70 WRKY TFs. At a minimum, all AtWRKYs in IIa should be included.  Also, the text for Figure 2A should be increased as it is currently illegible.

Ln164: Why name the transgenic lines 3, 7, and 13?  That might have been their lab-name, but in the publication why not refer to them as lines 1,2, and 3.  Further, throughout the manuscript no specific over-expression line is mentioned. Were all three characterized or just one?

Figure 3:  A) There is no way to distinguish WT from transgenic lines in panel A. Are all 3 transgenic lines included in the image? Please add labels. At 25uM Al, I don’t see any plants in panel A that look 30% shorter than the others; maybe at 50uM the first two look shorter?

Figure 3 and 4: How many plants of WT and each transgenic line were used for these statistics? Three of each line or three total? 

Ln 187: What is MDA and POD?

RNA-seq: On all three lines compared individually to WT? Or were all transgenic lines pooled together?  What genotype was used for WT; the tolerant or sensitive line?  Roots and leaves or just roots? 

            I’ve never done RNA-seq of transgenic plants, but if the transformation event wasn’t in the same place (a statistical improbability), this could result in the same phenotype, via different metabolic pathways.  Did the authors look at each line individually to see if there were differences between lines?  Even reading the materials and methods, I am unclear what plants, tissues, and how many biological replicates were used for the RNAseq experiment. There are also no RNAseq analysis parameters provided. What program were used?  Since GmWRKY81 is over-expressed, the authors should make sure the expression is highly up-regulated, then the gene should be masked from the GFF file and gene expression analyses should be run without it included.  This will make sure the extra reads associated with GmWRKY81 do not obscure or shift the analysis.  What comparisons were made?

            Were there plants under normal growth conditions and under Al stress conditions? Does GmWRKY81 change in response to Al stress? It doesn’t have to change to be a good candidate gene, but it’s an important thing to know and report.

Ln 224: How did you do the GO analysis?  For reporting GO results, you only need to pick one level – either Biological Process, Cellular components, or Molecular Function (I suggest biological process as it is usually the most informative).  I recommend using the Gene Ontology Overrepresentation tool on SoyBase, it was designed for the soybean genome. It also allows your results to be consistent with other soybean studies.

Ln 242: There should be citations for “Glyma.09G122600 and Glyma.20G02250 contribute to metal tolerance”.  That’s pretty specific, you must have a citation to back that up.

Ln 264: GmWRKY81 isn’t a candidate gene identified from RNAseq.  However, the role of GmWRKY81 in Al stress tolerance is being investigated using RNAseq. A small, but important difference.

Ln 284: WRKYs ARE transcription factors, so you can’t speculate that GmWRKY81 is acting as a transcription factor.

Ln 287: Research you cite shows soybean responds very quickly to abiotic stress. That GmWRKY81 isn’t changing in expression until 6 hours after stress is induced indicates it is not the primary response, but a down-stream response to the stress.  Does GmWRKY81 respond the same way to other abiotic stresses in soybean, or does the homolog respond similarly to other abiotic stresses in Arabiodpsis?   I think the majority of your results are a response to oxidative stress, but not necessarily Al toxicity; more general responses.

Ln 307:  This sentence is a mess.

Ln 316: Is Al stress alleviated by organic acids in soybean or any other species?  If so, there should be citations and this might be a valid sentence. As stated, it seems to be a hypothesis without much to back it up.  For example, does malate application alleviate Al stress?

Lns 324-329: Is Glyma.02G157900 differentially expressed in the experiment described by this manuscript? If yes, please add details of it’s expression and this discussion can remain. If not, then this seems a tangent that should be removed.

Ln 333: “distribution and transportation of metal ions”…. Does this include Al?  Specifically?  If yes, please say so, if not, this should be removed.

Reference 37 – I can’t find it online. I’m not confident that isn’t a predatory journal.

Reference 38 – this study is examining the role of root colonization with an endophytic fungus, which increases aluminum concentrations, but does not indicate that any of the genes enumerated in the manuscript (Glyma.06G115800, Glyma.08G175400, or Glyma.15G251300) are aluminum transporters

This paper is difficult to read because it’s lacking a lot of details. I looked at the supplementary data hoping for clarifications.  However, the DEG list doesn’t include GlymaIDs.  I assume this is because the authors used the genome sequence on NCBI rather than the genome sequence on Phytozome. Same genome, but different gene names.  Throughout the manuscript genes are referred to using their GlymaIDs. These Glyma IDs must be included in the differential gene list provided as Supplementary Table 2.  Without those IDs, Table S2 is effectively useless.  Additionally, there should be words, not abbreviations after the FPKM. No one outside the authors group will know what W100X and GXCK stand for.  Given that there is W100X_1 and W100X_2, does that mean two biological replicates or Wild Type and then two biological replicates of the transgenic line?  Again; the lack of details throughout the manuscript, but particularly in the materials and methods, leaves readers with more questions than answers.

Round 2

Reviewer 2 Report

There were so many changes to the manuscript that I found it difficult to read, but I do believe the authors have made every attempt to address the issues I pointed out in my original review.  I can tell this was a lot of work and it really improved the manuscript.  However, it's still very vague and unclear in places.

Items not addressed include:

1.  Does this study investigate Aluminum toxicity or deficiency?  The answer is revealed in the response to reviewers, but it's not obvious in the manuscript itself. 

2.  How big a problem is this for soybean?  Obviously in a lab setting soybean responds, but what is the yield loss for soybean every year due to Aluminum?  I don't think it's very high, which is one of the reasons I rated this study lower. 

3.  The authors provided a beautiful answers to points 7 and 16 in their response, but that information is not actually incorporated into the manuscript itself.  

4.  The figures and figure legends are still not properly aligned. 
